

# Brief communication: "Oldest Ice" patches diagnosed 37 km southwest of Dome C, East Antarctica

Olivier Passalacqua[1], Marie Cavitte[2,3], Olivier Gagliardini[1],
Fabien Gillet-Chaulet[1], Frédéric Parrenin[1], Catherine Ritz[1], and Duncan Young[2]

[1]Univ. Grenoble Alpes, CNRS, IRD, IGE, F-38000 Grenoble, France
[2]University of Texas Institute for Geophysics, Jackson School of Geosciences, University of Texas at Austin, Austin, Texas, USA
[3]Department of Geological Sciences, Jackson School of Geosciences, University of Texas at Austin, Austin, Texas, USA

*Correspondence to:* Olivier Passalacqua (olivier.passalacqua@univ-grenoble-alpes.fr)

**Abstract.** The presence of ice as old as $1.5\,\mathrm{Ma}$ is very likely southwest of Dome C, where a bedrock relief high makes the ice thin enough to prevent basal melting. Three-dimensional ice flow modelling is required to ensure that the basal ice is old enough above the bedrock, and that the age resolution of the ice archive is sufficient. A 3D ice flow simulation is led to calculate selection criteria that together

locate patches of ice with likely old, well-resolved, undisturbed and datable ice. These patches on the flanks of the bed relief are a balance trade-off between risks of basal melting and sufficient age resolution. The trajectories of the ice particles towards these sites are short and the ice flows over a smoothly undulating bed. Several precise locations of potential $1.5\,\mathrm{Ma}$-old ice are proposed, to nourish the collective thinking on the precise location of a future drill site.

## 1 Introduction

Antarctic ice is an exceptional archive of the Earth's paleoclimates across all the glacial/interglacial periods, and the only one that contains direct samples of ancient atmospheres. The oldest available ice archive goes back $0.8\,\mathrm{Ma}$ in time (EPICA Dome C ice core, Jouzel et al., 2007), but is not old enough to study a main climatic transition that occurred between $1.2\,\mathrm{Ma}$ and $0.9\,\mathrm{Ma}$, known from

the temporal variations of the isotopic composition of benthic sediments (mid-Pleistocene transition, MPT, Lisiecki and Raymo, 2005). The main climatic periodicity and the amplitude of climate cycles seem to radically change during the MPT. There is presently no general agreement on the processes responsible for this transition and its origin, and the influence of the trend of atmospheric $CO_2$, the dynamics of ice sheets in the northern hemisphere, the sea ice extent, or the dust content of the

atmosphere have been proposed (Clark et al., 2006). Considering these scientific issues, locating a future $1.5\,\mathrm{Ma}$-old ice drill site was identified as one of the main goal of the ice-core community (Jouzel and Masson-Delmotte, 2010).





Heat and mechanical ice flow simulations have to be carried out to assess the potential of possible drill sites (Fischer et al., 2013). The present study is part of a serie of three modelling exercises led by the IGE group and collaborators around Dome C, and more specifically on the bedrock relief lying $\sim 40\,\mathrm{km}$ southwest of Dome C, suspected to be a good old-ice candidate (Van Liefferinge and Pattyn, 2013), under the ice ridge linking Dome C to the Vostok region (for a precise description, see Young et al., 2017). First, we determined the energy balance of basal ice through the glacial-interglacial periods (Passalacqua et al., 2017), explaining the origin of the local spatial distribution of subglacial water at the ice-bedrock interface. The results of our transient simulations show that the top of this bedrock relief is likely to be melt-free on long timescales, or to undergo very limited basal melting. Hence, this subregion is thermally compatible with the archiving process on glacial/interglacial timescales. Second, we constrained a 1D mechanical model with dated ice-penetrating radar isochrones to evaluate the age and age resolution of the deepest portion of the ice sheet (Parrenin et al., 2017). This approach inverts the dated isochrones for a thinning parameter that characterizes the vertical deformation through the ice column. We showed that the observed isochrones are compatible with high basal age and sufficient resolution. However, these two previous studies neglect horizontal motion of ice, whereas the trajectories of the ice particles are definitely not vertical. The travel time of these particles, and consequently their age, is strongly influenced by the shape of the bedrock and the ice surface. We are presently lacking the description of the local 3D state of stress of the ice, where the geometry of the terrain and the strain history of the ice particles can be properly taken into account.

To evaluate the quality and position of deep old ice, we here proceed to a steady state 3D ice flow simulation, at a regional scale. Whereas these two previous 1D modelling results could not help us determine precise oldest-ice targets, we here intend to provide objective criteria that together delimit kilometer-scale patches of old, well-resolved, undisturbed basal ice. The bottom-most ice recovered should be older than the MPT, ideally as old as $1.5\,\mathrm{Ma}$ such that several climate cycles pre-MPT can be recorded. The vertical age resolution has to be better than $10\,\mathrm{ka\,m^{-1}}$ to detect high-frequency climatic variability in the ice core (Fischer et al., 2013). Moreover, basal ice will probably be disturbed, similarly to the deepest $60\,\mathrm{m}$ of the EPICA Dome C ice core (Jouzel et al., 2007; Tison et al., 2015). We have no better evaluation of the height of disturbed ice in the region, and we will similarly take $60\,\mathrm{m}$ as a security margin for the drilling minimum distance from the bedrock, but we should keep in mind this cutoff height could be an underestimation. One should look for places where the mechanisms responsible for stratigraphy disturbances (cumulated basal shear, bedrock roughness) should be minimal. Convergent flow should be avoided as well, because it tends to thicken basal layers. This is defavourable to recovering oldest ice as it will shift older layers downwards, and makes dating process more complex (Tison et al., 2015). Finally, location of the future drill site should be above the highest subglacial lake detected by the radar survey (Young et al., 2017), otherwise the risk of basal melting could be drastically increased ("water limit").



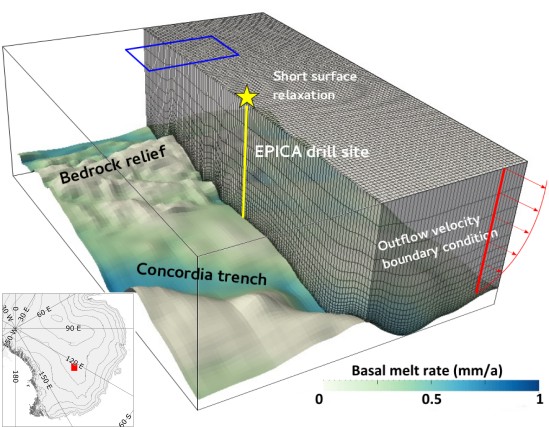

**Figure 1.** Mesh, bedrock dataset (Fretwell et al., 2013; Young et al., 2017) and basal melt rate (Passalacqua et al., 2017) used for the simulation. The red patch on the situation map (bottom left) show the hold of the domain used for calculation, and the blue outline corresponds to the hold of Fig 2.

## 2 Ice flow model

### 2.1 Model description

The Stokes equations are solved on a $83 \times 114$ km domain, approximately centered on the Dome C, using the finite element model Elmer/Ice. The surface and bed geometries are provided by the Bedmap2 data set (Fretwell et al., 2013), except on the bedrock relief southwest of Dome C, where a dense airborne radar survey has been recently collected (Young et al., 2017). The firn is accounted for by considering a mass-equivalent layer of ice at the surface; the modelled ice thickness is then simply 30 m thinner than the observed thickness (Schwander et al., 2001). Horizontal resolution of the corresponding mesh is 1 km. This mesh contains 20 vertical elements, the deepest one being 25 times finer than the upper one, so that the velocity field is better resolved near the bottom.

Our domain sits in the center of the Antarctic plateau, and lateral boundary conditions do not correspond to physical boundaries, but to virtual vertical surfaces. As the domain contains the dome summit, the ice flow is divergent and no input flow should be considered. We impose velocity boundary conditions corresponding to the shallow-ice approximation (SIA) (Hutter, 1983). The ice surface as observed today may not correspond to the chosen rheology and is relaxed for 50 years such that the present surface slope does not induce an unrealistic velocity field. The part of ice motion attributed to basal sliding is not known precisely in the Dome C region, and depends on water circulation. As we here focus on a region were basal melting is probably null, and for the sake of simplicity, a no-sliding condition is imposed at the bottom of the ice column. Vertical velocities at the base are equal to the basal melt rate output from previous modelling work (Fig. 7 in Passalacqua et al., 2017).





As we know that the basal ice in the Dome C region is at or near the melting point (Passalac-
qua et al., 2017; Van Liefferinge and Pattyn, 2013), the temperature profile measured in the EPICA
Dome C borehole is a good representation of the thermal structure of the ice in the Dome C vicin-
ity. Hence, we account for ice temperature by using the same normalized temperature profile on
all the domain. Solving the coupled thermo-mechanical equations would require heavy computing

resources, without radically changing the ice fluidity – which is mainly controlled by temperature –
and the trajectories of the ice particles.

    As the main interest of this work focuses on what happens for deepest ice, mechanical anisotropy
of the ice has to be accounted for. The relation between strain rates and stresses are described by the
generalized orthotropic linear flow law (GOLF, Gillet-Chaulet et al., 2005), given a certain vertical

profile of ice fabric. By introducing a dependence on the second invariant of the deviatoric stress, this
law can be extended to the non-linear case (Ma et al., 2010). The fabric profile is only known at the
dome summit (Durand et al., 2009), and shows that the ice mainly undergoes vertical compression,
but also longitudinal extension in the deep layers (Tison et al., 2015). However, there is no reason
to use the very same fabric profile everywhere else, where shearing is more influent or the bed

shape different. In this short study we will not discuss the influence of the chosen rheology, but we
first made sure that the computed surface velocities correctly simulated the horizontal velocity field
measured at the surface by Vittuari et al. (2004) for different $n$ values and fabric profiles. Hence,
we decided to use the widely-accepted value of 3 for $n$, and a synthetic vertical fabric profile, for
which the eigenvalues of the second order orientation tensor evolve linearly with normalized depth,

from isotropy at the surface to a single maximum fabric at the bottom (see a similar treatment in Sun
et al., 2014).

### 2.2   Model outputs

Back trajectories are computed from the 3D velocity field using a Lagrangian scheme such that the
age is known along the forward trajectory of the ice particles, and an age field is generated. The age

resolution could be calculated from the vertical derivative of this age field, but we found it more
accurate to track the annual layer thickness $\lambda$ from the ice surface (Whillans, 1979) using

$$\frac{d\lambda}{dt} = \lambda \, \dot{\epsilon}_{zz} \qquad\qquad\qquad (1)$$

where $\dot{\epsilon}_{zz}$ is the vertical strain rate. This formulation neglects vertical rotation effects that tends to
overturn internal layers. This assumption is reasonable if internal layers are mainly horizontal, which

is the case over the studied bedrock relief. The age resolution is then computed as the inverse of $\lambda$,
considered as the layer thickness for a single year.

    The way ice strains by flowing over a rough bed differs depending on the orientation of ice flow,
and a same bedrock relief could be a convergence or a divergence area. Once the velocity field



is known, a local coordinate system $(X, Y, Z)$ can be defined at each point, for which the $X$-axis is

oriented along flow. The curvature of the bed perpendicular to the flow is then computed everywhere, and convergence areas are identified where the bed curvature is positive.

Beyond the computation of ages and age resolutions, the 3D simulation is also useful to detect where deep ice is more likely to be folded. Shearing will tend to amplify small wrinkles in the ice layers and so disturb the ice's basal stratigraphy, whereas longitudinal extension will tend to flatten

them. Competition between shear and longitudinal stresses can be represented by a dimensionless shear number (Waddington et al., 2001)

$$\mathcal{S} = \frac{2\dot{\epsilon}_{XZ}}{\dot{\epsilon}_{XX} - \dot{\epsilon}_{ZZ}} \tag{2}$$

where $\dot{\epsilon}_{XZ}$ is the shear strain rates along ice flow, $\dot{\epsilon}_{XX}$ and $\dot{\epsilon}_{ZZ}$, the local longitudinal and vertical strain rates. Waddington et al. (2001) use this shear number as a criterium to detect if a given wrinkle

can be amplified by shear. More simply here, we use it to predict the presence of undisturbed ice, whereby the smallest shear number is best.

### 2.3 Selection of favourable locations

If the absolute value of the age, age resolution, or strain rates can be discussed regarding the choices of the model parameters, the outputs still keep their relevance when analysed relatively to them-

selves. The existence of older, better-resolved, less disturbed ice, is much less sensitive to the magnitude of ice velocity than to the shape of the bedrock or that of the ice surface elevation. As a consequence, this short study focuses on getting practical information for the decision-making process rather than on discussing the uncertainties or the influence of the chosen model parameters.

The five selection criteria (age, age resolution, bed curvature, shear number and bed height) are

used to compute five masks, thresholded as follows. For bed curvature, $0 \, \mathrm{m}^{-1}$ would have been the natural threshold but it was too much restrictive and we decided for a slightly higher value $(2 \, 10^{-5} \, \mathrm{m}^{-1})$. The shear number threshold appeared naturally by studying its spatial evolution (see §3.3). The bed height threshold correspond to the "water limit", at $480 \, \mathrm{m}$. Furthermore, our results show that most of the subglacial elevated bed relief southwest of Dome C is favourable to the ex-

istence of $1.5 \, \mathrm{Ma}$-old ice, so we adopted more conservative age and age resolution thresholds for the selection of smaller suitable locations ($1.8 \, \mathrm{Ma}$ for the age and $8.5 \, \mathrm{ka} \, \mathrm{m}^{-1}$ for the age resolution, $60 \, \mathrm{m}$ above the bed). A logic combination of the five masks delineate the patches fulfilling all our selection criteria.



## 3  Results and interpretation

### 3.1  Age at 60 m above the bed

The area identified as possibly hosting oldest ice is elongated along the bedrock relief, and stands at an intermediate bed elevation (mainly between 400 m and 550 m above seal level, Fig. 2). Neither the very top of the bedrock relief, nor its lowest foothills appear to be suitable for the archiving process of very old ice. The basal melt rates imposed as boundary condition are null on the upper part of the bed relief, therefore infinite age are calculated for the very basal ice. In that case, the age of the ice standing 60 m above the bed is strongly dependent on the ice thickness. On the top of the bedrock relief, the ice is at its thinnest and the old ice then sits closer than 60 m above the bed. On the foothills of the bedrock relief, the ice is thick, basal melt rates are above zero, and the basal ice is therefore continuously being melted from the bottom.

Some places may host ice even older than 2 Ma, but they all stand below Young et al's (2017) water limit (Fig. 2, yellow line). The presence of very old ice in those areas is not impossible, but may also be the consequence of insufficient imposed basal melting. The transition between melting and frozen ice should stand somewhere on the flanks of the bed relief, but it is difficult to pinpoint the precise location of this threshold. Despite the promising thick ice in the region to ensure old ages and sufficient age resolution, the risk of basal melting is real.

### 3.2  Age resolution at 1.5 Ma

Age resolution for the deepest part of the ice column is influenced by two factors. First and obviously, the thicker the ice, the better the age resolution. As a consequence, the tops of the bedrock relief are rarely compatible with a sufficient age resolution of oldest ice. Bedrock flanks should be preferred, but some of the thickest ice areas on the flanks will be discarded as well because of an increased risk of basal melting.

Second, for ice positioned close to the ice ridge, age resolution benefits from a thickening effect of the deeper layers (so-called Raymond effect, Raymond, 1983). This results in a band of well-resolved ice, oriented along the ice ridge and perpendicular to the bedrock relief. No Raymond arche is visible in the radargramms that could argue for a strong Raymond effect time here. One explanation is that the shape of the ice surface is much more rounded than at Dome C, and the produced along-ridge flow tend to dampen the amplitude of the Raymond arches (Martín et al., 2009). Moreover, the characteristic time for a Raymond arche to form here would be several 100 ka (Martín et al., 2009), during which the surface ridge probably moved, smoothing out the forming arches. Unfortunately, the past position and shape of the ridge are unknown, and drilling far from its present position would not guarantee a better resolution.





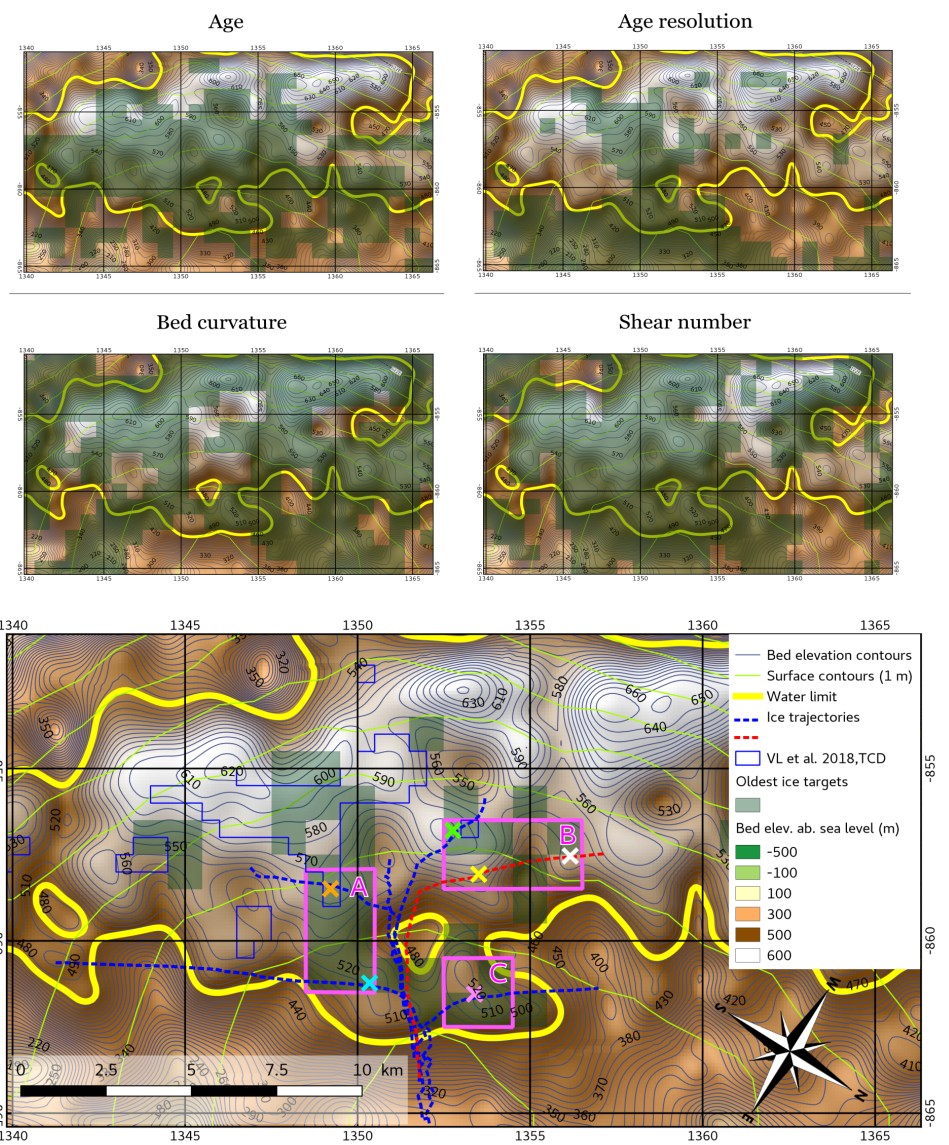

**Figure 2.** This figure focuses on the bedrock relief designed as such in Fig.1, located $\sim 40\,\mathrm{km}$ southwest of Dome C. The upper, small maps present the selection criteria thresholded as following : age $> 1.8\,\mathrm{Ma}$, age resolution $< 8.5\,\mathrm{ka\,m^{-1}}$, bed curvature $< 2\;10^{-5}\,\mathrm{m^{-1}}$, shear number $\mathcal{S} < 40$, and bed height $> 480\,\mathrm{m}$ (simply shown here by a thick yellow contour). The bottom map shows the combination of the 5 selection criteria. Magenta boxes A, B and C correspond to patches that could be considered as our best oldest-ice targets. Dashed lines show trajectories of ice particles, the red one correspond to the profile presented in Fig. 3. Crosses locate possible drill sites, discussed in the text. Blue outlines show the best patches of Van Liefferinge et al. (2018). Projection: WGS84/Antarctic Polar Stereographic – EPSG:3031 (kilometers).



### 3.3 Stratigraphic disturbances

At a divide, the shear stress perpendicular to the divide is null, so that the shear number $\mathcal{S}$ of the deep ice close to the ridge is low ($\sim 10$). However, ice flow faces steep bed slopes with higher

velocities than under the divide west side of the bedrock relief. As a consequence the shear number increases very sharply and reaches much higher values ($\sim 100$), and this zone of high shear should be discarded for the oldest-ice challenge. All the areas for which the basal ice crossed this zone of high shear should be discarded as well, so the trajectories of the ice particles need to be represented.

### 3.4 Trajectories for oldest-ice spots

The best combination of age, age resolution, folding, convergence and melting criteria is shown in Fig. 2 (bottom), revealing several spots of appropriate ice. The patches for which the trajectories are the shorter should be preferred, and two magenta boxes highlight our most promising patches. For magenta boxes A, B and C the ice originates from the divide, guided by the strong lateral divergence resulting from the shape of the ice surface. Locations within boxes C and A should be considered

first for a future Oldest Ice drilling because of shorter trajectories, and less risks of stratigraphy disturbances. But in the box B stands a relatively flat bedrock platform (white cross in Fig. 2, bottom), that could ensure a certain stability of the flow. Figure 3 shows the travelling of the ice particles towards this site, and the corresponding horizontal distance travelled does not exceed 10 km from the surface. However, as there is probably no basal melting here, the ice particle would be likely to

closely follow the bed along several kilometers, in a depth range dominated by a strong vertical shear, enhanced by an unduling bed underneath. To minimize the bed influence, we could also consider a drill site located 3 km upstream, where the ice age would exceed 1.5 Ma (yellow cross in Fig. 2, red dashed line in Fig. 3).

Of course, locating a unique "best" drill site within one of these three boxes is not possible with

this 3D-modelling approach only. However, it allows to define a restricted area where a new set of observations will be the most valuable. We should focus on local bedrock summits or crest lines, because local troughs make the ice flow converge, but also heat flow (Van der Veen et al., 2007), increasing the risk of insufficient age and positive basal melt. Furthermore, the ice ridge probably moved laterally in the past, which is not accounted for here. As a consequence the true trajectory

of an ice particle for a given drill site could even come from the northwest and turn left, or from the southeast and turn right. By choosing a local bedrock summit, overhanging its environment, we minimize the risk that the true trajectory of basal ice met a bedrock obstacle. Considering that, only a few set of favourable drill sites remain in boxes A, B and C (crosses in Fig 2).




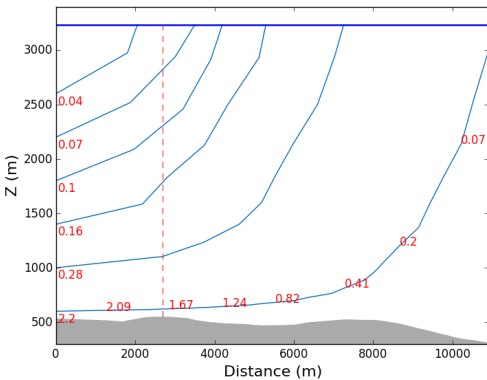

**Figure 3.** Trajectories of the ice particles from the ice surface towards the bedrock (red trajectory in Fig. 2). Red numbers indicate the age of the ice, in Ma. The bed shape is shown in grey. The thin red dashed line show one possible drill site (yellow cross in Fig. 2).

### 3.5 Comparison with results based on thermodynamical modelling

Our results can be benchmarked against the ones of Van Liefferinge et al. (2018), who used a transient thermodynamical model to compute the highest geothermal flux value that keep basal ice frozen. By comparison with available continental geothermal flux datasets, they locate patches where basal ice has likely remained frozen over 1.5 Ma. These authors also included a further mechanical constrain representing limited impact of bedrock roughness on the preservation of the bottom ice

stratigraphy. They identified a 8 km-long patch covering the SE upper part of the bedrock relief, that crosses our biggest, central patch, but both do not overlap pefectly (Fig. 2, bottom map, blue outlines and green areas). They also identified a site within our magenta box B, but no site in the box C. These comparative results highlights the complementarity of the two approaches. The 1D thermodynamical model of Van Liefferinge et al. (2018) has a better control on the thermal aspect of the problem

than on its mechanical aspect, and selects sites that are more conservative from a heat budget point of view, i.e. preferentially local bedrock heights. On the contrary, our 3D approach accounts for the horizontal strain of the ice, and select sites that are more conservative from a mechanical point of view. The upper part of the box A or the left part of the box B validate the constraints of both approaches. In our approach, the bedrock summit in box C is the safest mechanically; however, it was

not selected by Van Liefferinge et al. (2018) because of a local bedrock roughness exceeding their threshold of 20 m, despite the fact that their thermal criterium was fulfilled.





## 4 Conclusions

The three-dimensional ice flow simulation presented here aims at defining and calculating several objective criteria, which represent ideal conditions for the retrieval of old, well-resolved, undisturbed ice. The influence of the bedrock relief and of the position of the ice ridge allows us to define only a few patches compatible with all our selection criteria. The ages calculated at the base by our model simulations predict ice older than $1.5\,\mathrm{Ma}$ old high enough above the bedrock, which gives us confidence that the community's target of $1.5\,\mathrm{Ma}$ years should be attainable, with the required age resolution. However, the modelling approach implicitly assumes that the ice flow is regular down to the bedrock, but there is no guarantee it is actually the case. A ground radar survey focusing on the kilometer-scale patches presented here is essential to explore the structure of the deep layers. Finally, a rapid-access drill (Grilli et al., 2014) is currently planned to be deployed in the 2018/19 season to assess ice quality and age for a chosen target site, before the final site is decided upon.

*Acknowledgements.* The Australian Antarctic Division provided funding and logistical support (AAS 3103, 4077, 4346). This work was supported by the Australian Government Cooperative Research Centres Programme through the Antarctic Climate and Ecosystems Cooperative Research Centre (ACE CRC); support for UTIG came from the G. Unger Vetlesen Foundation and NSF grant PLR-1443690. This paper is UTIG contribution number ... This publication was generated in the frame of Beyond EPICA-Oldest Ice (BE-OI). The project has received funding from the European Union's Horizon 2020 research and innovation programme under grant agreement No. 730258 (BE-OI CSA). It has received funding from the Swiss State Secretariate for Education, Research and Innovation (SERI) under contract number 16.0144. It is furthermore supported by national partners and funding agencies in Belgium, Denmark, France, Germany, Italy, Norway, Sweden, Switzerland, The Netherlands and the United Kingdom. Logistic support is mainly provided by AWI, BAS, ENEA and IPEV. The opinions expressed and arguments employed herein do not necessarily reflect the official views of the European Union funding agency, the Swiss Government or other national funding bodies. This is BE-OI publication number ... This publication also benefitted from support by the Université Grenoble Alpes in the framework of the proposal called Grenoble Innovation Recherche AGIR. This work was granted access to the HPC resources of CINES under the allocation A0020106066 made by GENCI. We thank Brice Van Liefferinge for making its datasets available.




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
