# Peer review of "Brief communication: Candidate sites of 1.5-Ma ice 37 km southwest of the Dome C summit, East Antarctica"

_The Cryosphere, 2018_

## Referee Comment (RC1) · M. Koutnik (Referee) · 21 Mar 2018

The authors present a new modeling study that further characterizes the ice-flow and subglacial conditions that could combine to best preserve ice up to 1.5 million years old near Dome C. The study builds on previous work by applying a 3-D model and by using the latest available bed topography. The authors identify candidate drill sites that attempt to balance needs of the oldest ice possible with the age-resolution desired for ice-core analyses.

This was a nice piece of work that advances understanding of the Dome C environment; modeling like this is key to picking a drill site for the "Oldest Ice" ice-coring effort.

I have more substantive comments on the style of the manuscript, and relatively minor comments on the modeling. Both are given below, and by line number as appropriate.

I suggest that the authors work together to improve the writing style (including title, abstract, main text, and summary statement). The manuscript can be understood but the language used is often non-descript, and I think detracts from the impact of the work. Sometimes it is a subtle use of an inappropriate word, and sometimes (at least to me) it gives a context to the sentence that may or may not be intended. I appreciate that this may be a simple translation situation – and in my attempts to learn French I am very sympathetic to how difficult this could be – but, again, I think it is worth rewriting this carefully using different expressions so that the work is clear and has the most impact for all readers. Hopefully all of the authors can work together to achieve this.

Line 2: "prevent basal melting" – suggest "limit"

Line 3: "ensure" is strong, really you are making the best estimate

Line 4: "ice archive is sufficient" – sufficient for what? It hasn't been made clear what is needed, there is a disconnect between first sentence and following sentences

Line 5, title, and throughout: I am not against the use of the term "patches", but the term doesn't tell the reader much. Since this is in support of ice coring, one impression is that a patch could be a few meters, or since the study area around Dome C is tens of km+ maybe this is the scale? I would clarify what area a patch covers up front, and I also suggest making a somewhat more general title (why is 37 km so important?). I think "area" is a better term than "patches", but the authors can justify what is best based on use in the community.

Title: "diagnosed" is not incorrect, but isn't the way I would expect it to be used. None of these have to be used, but some title suggestions could be: "Candidate areas of 1.5-Ma ice southwest of Dome C, East Antarctica", "Flow-model constraints on locations where 1.5-Ma ice exists southwest of Dome C, East Antarctica", … something more
general, and yet specific to the needs of "oldest ice" seems better.

The stated challenge with this is that the model is only as good as the boundary conditions that may vary in space and/or time. I understand the need to say something strong about the presence of 1.5-Ma ice so that the next steps toward a drilling program can proceed. I think the authors acknowledge this but I'm left with the tension on whether it is better to state the results more confidently, or less. Perhaps this is where language comes in again, for example:

Lines 9-10: "Several precise locations of potential 1.5-Ma-old ice are proposed, to nourish the collective thinking on the precise location of a future drill site."

I suggest that this sentence is revised to state directly what the community should do with these results – or, is already doing with them! If the results are really just something to think about, I guess that is it. But, if they are the state-of-the-art in modeling and what will in fact be used going forward as a community, say that. (And, "nourish" isn't the right usage here and gives too loose a sense of the value of this work.) Also, from the conclusions it sounds like this modeling has informed where to collect new radar data, right?

Also, why is this a "Brief communication"? It seems awkward for a modeling paper that should contain enough detail to evaluate the merits of putting results to use in planning would be published as a "brief". Again, this was hard to evaluate because it wasn't entirely clear how the results from this work should be used. Are they really just something to think about as the other work on this moves forward? Why is this a valuable step? I think that the results will in fact be used more directly hand-in-hand with new data collection, rapid-access drilling, and eventual deep drilling – again, I suggest the authors frame their results more directly in context with the community effort.

Line 17: Might want to elaborate on whether processes under consideration are external, internal, or both. And, seems that there would be more references other than

[Figure]

Clark et al. (2006), so give as "e.g., " or list a few more that are relevant – possibly splitting out after points in this sentence where the references apply.

Line 25: What is "IGE"?

Line 35: "inverts" is not used correctly here, and I would state more directly that this approach solves an inverse problem, making clear the model parameters that are inferred

Line 37: "definitely not vertical" – suggest as "... are not only vertical"

Line 53: "security margin" – suggest other phrasing, and while I understand there is no better estimate, is that really true? What about ice-flow conditions between candidate sites in this work and Dome C drill site may inform if 60 meters is an under- or overestimate? Did you try other values if there is a chance this is an underestimate (as stated)? Line 56: "defavourable" is not a word, here it would be "unfavourable"

Line 58: missing "the" between "Finally, the location..."

Line 59: Define the "water limit", I think I understand but since it is used often need to be clear what this is and how it is estimated

Figure 1: A scale bar would be helpful

Figure 1 caption: I would refer to this a "context map" instead of a "situation map". Language of "the hold of the domain" sounds off, and I suggest "... shows the domain used for the calculation, and the blue rectangle at the top of the image is the location of Figure 2"

Line 61: Refer to Figure 1?

Line 66: Is the firn really "accounted" for? I would say that your model is in ice equivalent and you adjust the surface height using an assumed density profile to convert the firn layer to an ice layer. It should be clear that you don't include a process model of firn. Where did your density profile come from? (Assume Dome C, but did you apply

that everywhere?)

Does the model resolution as a function of depth vary linearly, exponentially, ?

Line 77: "were" should be "where" I suggest using "not present" instead of "null"

Line 84: Instead of "heavy", suggest "excessive" – and is that really true for the limited domain of your model given that it is steady state? Is the issue that you can't solve the time-dependent problem and therefore a coupled thermomechanical model doesn't add much in steady state?

Also, it could be worth noting that it is non-trivial to extend a multi-dimensional limited-domain model from a steady-state calculation to a transient one. Without modeling the full continent you need to impart information to this regional model about how ice-thickness changes and ice-flow changes inside and outside of this domain correspond to changes in the rest of the ice sheet in which it is embedded. So, for the goal of regional modeling you are minimizing even more assumptions (and challenges in setup and computation) by starting in steady state.

I looked at this for 2.5-D (flowband) models: Koutnik and Waddington (2012), Well-posed boundary conditions for limited-domain models of transient ice flow near an ice divide, Journal of Glaciology 58, 1008-1020.

Line 94: Instead of "more influent", suggest "has more influence"

Line 110: State as 1/lambda?

Line 113-114: This sentence wasn't clear to me starting with "The way ice strains..."

Line 143: What do you mean by "a logic combination"?

Figure 2: x-axis and y-axis numeric labels are way too small in the top panels, and probably also too small in the bottom panel. Axis labels are missing. I am not really sure where to read the numbers from each panel and without those they don't say much – the caption needs to be improved to make sense of these panels, or maybe

[Figure]

the top four are not shown? It looks like there would be more overlapping areas, or is it just that age resolution is limited?

Make sure the caption clearly guides the reader through this and that all box colors are identifiable. For example, it took me awhile to see the blue box showing location of Van Liefferinge results.

Seems like colored crosses should receive more discussion in the text. Pros / cons of each choice could be takeaway points. Again, how should the community use these results?

Putting these results in better context with the Van Liefferinge et al. (2018) results seems necessary. Especially that this work is still in The Cryopshere Discussion, the reader is not necessarily sure what to make of these parallel efforts. I might have missed it, but I think the first citation of this work is in the caption of Figure 2. Section 3.5 goes into more depth between the approaches but could be worthwhile to put this context up front, and as part of the framing of why your results matter to the community. Is there really no way and/or no effort underway to combine these two approaches? Is the limitation only computational? At what point might this be possible? (Or, what are the next steps that can be taken by the modeling community in this effort to find the best drill site?)

Line 182: I suggest rephrasing "oldest-ice challenge" – even "the challenge of finding the oldest ice" sounds better, somehow the other sounds too loose, and therefore does not have as much impact.

Line 187: "appropriate ice", suggest rephrasing what you mean by "appropriate" or using the phrase "candidate drill sites"

Line 190: suggest using another word than "risks" (there are other usages that I'd also change"; maybe the word is "chance"

Line 190: By shorter trajectories you really just want to be closest to the dome / ridgeline

Figure 3: Why not give the x-axis in km? All labels are very small

Is the bed topography the most important boundary condition in your modeling? What resolution would be ideal to be confident from modeling on where to pick a drill location?

Line 210: How can these results really be benchmarked against Van Liefferinge et al. (2018) given that they use a very different approach? All you have done is qualitative comparison, right?

Line 214: should be "constraint"

---

## Referee Comment (RC2) · T. van Ommen (Referee) · 29 Mar 2018

This manuscript uses a three dimensional ice sheet modelling approach to explore the basal age and resolution of the ice sheet at Dome C, in a region where bedrock relief is likely to be conducive to preservation of very old ice: up to 1.5 Ma or more. The study appears well-posed and the work is a nice summation of what this approach can tell us regarding basal age and resolution. It is an important advance that is required for targeting future drilling locations and should be valuable in guiding additional exploratory studies. I have only minor comment concerning the modelling itself. The paper does suffer in places from somewhat non-standard English usage, some of which intrudes a

little on readability. While fully appreciative of the authors' first language I respectfully suggest these items be edited for clarity – noting a native English speaking author is on the list. I see the other referee has commented in similar vein – I will not generally specify the linguistic items for correction below. Detailed comments: Title: In general I disfavour the usage of Dome C as a "point location" synonymous with Concordia Station, as the entire region is really Dome C. Suggested use would be to have the title read "Oldest Ice" patches diagnosed at Dome C, 37 km southwest of Concordia Station. I however leave it to the authors to consider, as it is not a substantive concern.

Line 25: Define IGE on first use Line 37: " and [provide] sufficient resolution" Line 48: Fischer et al. actually stipulate no more than 20 ka m-1 although this may now be thought too coarse. I have heard targets of 14 ka m-1 used. In any case, the 10 ka m-1 is not consistent with the reference. Figure 1 caption: "show the hold [sic] of the domain" Line 74: relaxed for 50 years …. How is the reader assured that this is adequate? Naively it seems very short. Maybe just reword to say that this period proves sufficient to propagate away initial discontinuities or similar. Line 77: "focus on a region where basal melting is probably null" - this may be true for the high points, but the domain most certainly includes areas of basal melt, so how is it that a no sliding condition is OK? Line 95-100: It is not clear from the description why the use of stress exponent n=3 is valid. Indeed there is some varied opinion in the literature over the best value to use in various situations particularly ice divides (see e.g. Martin et al., JGR, 2009; Martin and Gudmundsson,TC, 2012; Petit et al., JGlac, 2007). While not wishing to create imbalance in the treatment in this paper by opening an extensive discussion, some context to the literature would be useful. More importantly for understanding the results of this modelling, could the authors arrive at a statement as to whether an exponent n=3 is likely to under- or over-estimate age and resolution? That is, is it conservative to the aim of finding old ice? Lines 128-130: Maybe an example of the language clarity issue, but it is hard to see what is meant by "the outputs still keep their relevance when analysed relatively to themselves" Line 138: the "water limit" at 480m needs a little explanation, where does it come from and what is the reference height (I

assume it means 480 m.a.s.l.). Line 171: An example where the "Dome C" not equal to "Concordia" nomenclature issue comes up. I'd favour "Concordia". Figures 2 and 3: Axis labels in particular are too small. Figure 3 would benefit from all text being larger. Line 216: "our biggest central patch" isn't so easy to follow as using the labels provided: I assume it is "Patch A".
* * *

---

## Referee Comment (RC3) · B. Hubbard (Referee) · 30 Mar 2018

The manuscript presents a potentially valuable 3D modelling-based study designed to isolate, through a series of spatial masks, candidate locations for "oldest ice" (1.5 Ma) near Dome C, East Antarctica. The analysis is interesting and, I believe, robust (subject to some reservations, below) and I would support publication. I believe the manuscript structure and approach are valid, but I do see the manuscript's current findings as somewhat undermined by the handling of temperature – and particularly basal temperature and sliding - in the analysis. I would encourage a revised manuscript to consider this in more detail, at least placing some first order approximations of error

based on possible temperature scenarios. I accept that this may not be the forum for a full thermo-mechanical analysis but, for the analysis is fit for purpose, it needs to report some approximation of the age errors that might derive from the assumptions made.

The writing is occasionally ambiguous and includes grammatical and typographical errors.

Some more specific comments follow:

Location/Line Comment/Suggestion

Abstract Some of the wording could be improved here between lines 3 and 7.

Page 2 There are at least two typographical errors on this page ("serie" and "de-favourable"). The manuscript needs checking to remove other occurrences. I identify a few more below.

30-31 Is it not possible that changes in ice thickness have an influence here at long timescales? Can this influence and outcome be approximated?

30-42 Can the simulations and models of others publishing on this topic be summarised briefly?

77-80 This reads as contradictory, including a statement that "basal melting is probably null" (I would use zero rather than null) and "Vertical velocities. . . are equal to the basal melt rate output from previous modelling. . .". I think the manuscript would benefit from an explanation and statement of possible error (in using this basal temperature field) here.

94 "has more influence"

124 "used"

128-130 This sentence is unclear

132-133 I do not believe this is a valid argument: either the analysis is fit for purpose or

it is not. For the former to hold then errors need to be constrained. If, as a consequence of the 'practicable' analysis undertaken, errors are large then the manuscript would benefit from those large errors being stated as usefully as possible.

158-159 Presumably reflected radar power could inform as to the current location of this boundary. Do such data exist and/or has such an interpretation been published elsewhere?

170 "arch" and "radargrams"

Figure 2 Given the importance of these domains I find that the green on brown shading isn't working very well. In fact, this figure could be improved in several ways including: formally numbering panels a-e; increasing axis label font size in a-d; changing the depiction of the oldest ice targets considering the underlying green bed elevation band; and rewording the first line of the caption to a more standard format.

183 I wonder why this is "discarded" rather than included as a mask in the way that other spatially-distributed variables are?

186 This reference to "bottom" highlights the need for panel labelling in the figure

200 "However, it allows a restricted area to be defined where..."

206 I believe "overhanging its environment" is inaccurate and does not convey the intended meaning. I think this argument needs to be clarified and formalized.

Figure 3 caption "shows"

221 I would replace "On the contrary" with "In contrast"

223 Could the references here to "upper part" and "left part" be replaced with compass directions north and west?

226 I'm not familiar with the third from final word in the sentence.

---

## Author Comment (AC1) · 18 May 2018

The authors present a new modeling study that further characterizes the ice-flow and subglacial conditions that could combine to best preserve ice up to 1.5 million years old near Dome C. The study builds on previous work by applying a 3-D model and by using the latest available bed topography. The authors identify candidate drill sites that attempt to balance needs of the oldest ice possible with the age-resolution desired for ice-core analyses. This was a nice piece of work that advances understanding of the Dome C environment; modeling like this is key to picking a drill site for the 'Oldest Ice' ice-coring effort.

[Figure]

I have more substantive comments on the style of the manuscript, and relatively minor-comments on the modeling. Both are given below, and by line number as appropriate. I suggest that the authors work together to improve the writing style (including title, abstract, main text, and summary statement). The manuscript can be understood but the language used is often non-descript, and I think detracts from the impact of the work. Sometimes it is a subtle use of an inappropriate word, and sometimes (at least to me) it gives a context to the sentence that may or may not be intended. I appreciate that this may be a simple translation situation – and in my attempts to learn French I am very sympathetic to how difficult this could be – but, again, I think it is worth rewriting this carefully using different expressions so that the work is clear and has the most impact for all readers. Hopefully all of the authors can work together to achieve this.

We would like to thank Michelle Koutnik for her fruitfull comments, that helped improve this manuscript. English wording was carefully corrected in this new version.

**Line 2: 'prevent basal melting' – suggest 'limit'**

L4: Changed for "limit"

**Line 3: 'ensure' is strong, really you are making the best estimate**

L5: Changed for the following sentence: "A 3D ice flow simulation is used to calculate five selection criteria, which spatial variability is used to locate areas that have better glaciological properties than elsewhere."

**Line 4: 'ice archive is sufficient' – sufficient for what? It hasn't been made clear what is needed, there is a disconnect between first sentence and following sentences.**

"Sufficient" is deleted in the new formulation (see previous answer).

**Line 5, title, and throughout: I am not against the use of the term 'patches', but the term doesn't tell the reader much. Since this is in support of ice coring, one impression is that a patch could be a few meters, or since the study area around Dome C is tens of km+ maybe this is the scale? I would clarify what area a patch covers up front, and**

[Figure]

I also suggest making a somewhat more general title (why is 37 km so important?). I think 'area' is a better term than 'patches', but the authors can justify what is best based on use in the community.

As we finally suggest precise drill points within kilometer "patches", we change the word for "sites". We think giving the distance from Dome C is important to inform the community that the research of a drill site is now focusing on a kilometer-scale region, and gives the location in a region that lacks place names.

**Title: 'diagnosed' is not incorrect, but isn't the way I would expect it to be used. None of these have to be used, but some title suggestions could be: 'Candidate areas of 1.5-Ma ice southwest of Dome C, East Antarctica', 'Flow-model constraints on locations where 1.5-Ma ice exists southwest of Dome C, East Antarctica', . . . something more general, and yet specific to the needs of 'oldest ice' seems better. The stated challenge with this is that the model is only as good as the boundary conditions that may vary in space and/or time. I understand the need to say something strong about the presence of 1.5-Ma ice so that the next steps toward a drilling program can proceed. I think the authors acknowledge this but I'm left with the tension on whether it is better to state the results more confidently, or less. Perhaps this is where language comes in again, for example:**

We accept your suggestion, and changed the title for "Candidate sites of 1.5-Ma ice 37 km southwest of the Dome C summit, East Antarctica". We removed the word "diagnosed" that complexified the title.

**Lines 9-10: 'Several precise locations of potential 1.5-Ma-old ice are proposed, to nourish the collective thinking on the precise location of a future drill site.' I suggest that this sentence is revised to state directly what the community should do with these results – or, is already doing with them! If the results are really just something to think about, I guess that is it. But, if they are the state-of-the-art in modeling and what will in fact be used going forward as a community, say that. (And, 'nourish' isn't the right**

usage here and gives too loose a sense of the value of this work.) Also, from the conclusions it sounds like this modeling has informed where to collect new radar data, right?

L10: We changed the end of the sentence to be more specific and to link our results to the ongoing field work: "These sites will help to choose where new dense ground radar surveys should be conducted in upcoming field seasons. ".

**Also, why is this a 'Brief communication'? It seems awkward for a modeling paper that should contain enough detail to evaluate the merits of putting results to use in planning would be published as a 'brief'. Again, this was hard to evaluate because it wasn't entirely clear how the results from this work should be used. Are they really just something to think about as the other work on this moves forward? Why is this a valuable step? I think that the results will in fact be used more directly hand-in-hand with new data collection, rapid-access drilling, and eventual deep drilling – again, I suggest the authors frame their results more directly in context with the community effort.**

We choose to publish our results as a brief communication for two reasons. First, we wanted to shed light on the practical side of our results, that may interest many different persons, mainly researchers that are not ice-flow modellers. A brief communication format is more accessible in this perspective. Second, our modelling work has several limitations, the main one being that we did not made any sensitivity study on the input parameters (lack of time and resources), that would have probably been asked for within a longer article. So we choose to present our results as they are now, and as they are actually used. Finally, the editor agreed that this format fitted our message.

**Line 17: Might want to elaborate on whether processes under consideration are external, internal, or both. And, seems that there would be more references other than Clark et al. (2006), so give as 'e.g., ' or list a few more that are relevant – possibly splitting out after points in this sentence where the references apply.**

As we are limited in space and number of reference, we specify to refer to Jouzel and Masson-Delmotte for a more complete overview of the problem.

**Line 25: What is 'IGE'?**

L28: The acronym is developped: Institut des Géosciences de l'Environnement (Grenoble).

**Line 35: 'inverts' is not used correctly here, and I would state more directly that this approach solves an inverse problem, making clear the model parameters that are inferred**

L39: The new sentence is now: "The distance between the dated isochrones and the modelled ones was minimized to infer a thinning parameter that characterizes the vertical deformation through the ice column".

**Line 37: 'definitely not vertical' – suggest as '. . . are not only vertical'**

L43:Changed for: "the trajectories of the ice particles are not only vertical."

**Line 53: 'security margin' – suggest other phrasing, and while I understand there is no better estimate, is that really true? What about ice-flow conditions between candidate sites in this work and Dome C drill site may inform if 60 meters is an under- or overestimate? Did you try other values if there is a chance this is an underestimate (as stated)?**

L59: We changed for "safety distance". The origin of the ice disturbance in the last 60 m at Dome C is not clear, and the ice layering of very deep ice (bed+200 m more or less) cannot be unambiguously interpreted from the radargramms. So it is difficult to constrain the spatial evolution of this disturbed layer. However, we show later in the paper that 1,5 Ma should stand higher than 60 m above the bedrock, so that this threshold does not prevent us from selecting the best sites.

**Line 56: 'defavourable' is not a word, here it would be 'unfavourable'**

L63: Changed for "unfavourable"

**Line 58: missing 'the' between 'Finally, the location. . .'**

L64: Missing word added

**Line 59: Define the 'water limit', I think I understand but since it is used often need to be clear what this is and how it is estimated**

L66: We added the following sentence: "We will call this threshold water limit, above which there is no evidence of the presence of water in the radargramms"

**Figure 1: A scale bar would be helpful**

A scale bar is not compatible with this oblique projection. So we added information in the caption: "Mesh, bedrock dataset and basal melt rate used for the simulation on a 83 x 114 km domain."

**Figure 1 caption: I would refer to this a 'context map' instead of a 'situation map'. Language of 'the hold of the domain' sounds off, and I suggest '. . . shows the domain used for the calculation, and the blue rectangle at the top of the image is the location of Figure 2' The caption is modified as follows: "The red patch on the context map (bottom left) shows the domain used for the calculation, and the blue rectangle at the top of the image is the location of Fig2."**

**Line 61: Refer to Figure 1? L86:Reference to Fig. 1 added.**

**Line 66: Is the firn really 'accounted' for? I would say that your model is in ice equivalent and you adjust the surface height using an assumed density profile to convert the firn layer to an ice layer. It should be clear that you don't include a process model of firn. Where did your density profile come from? (Assume Dome C, but did you apply that everywhere?)**

L88: Right, we have no firn model, only an ice-equivalent layer. The sentence is restated: "The model works in ice-equivalent, and we adjust the surface height by assuming that the density profile of the firn is the one of Dome C on the whole domain."

**Does the model resolution as a function of depth vary linearly, exponentially, ?**

L91: It evolves linearly, we mentionned it as follows: "The resolution of the 20 vertical elements of the mesh evolves linearly, so that the deepest one being 25 times finer than the upper one"

**Line 77: 'were' should be 'where' I suggest using 'not present' instead of 'null'**

L103:The new sentence is now: "We here focus on a region where basal melting is probably not present or limited, and horizontal velocities are very small, so that, for the sake of simplicity, a no-sliding condition is imposed at the bottom of the ice column."

**Line 84: Instead of 'heavy', suggest 'excessive' – and is that really true for the limited domain of your model given that it is steady state? Is the issue that you can't solve the time-dependent problem and therefore a coupled thermomechanical model doesn't add much in steady state? Also, it could be worth noting that it is non-trivial to extend a multi-dimensional limited-domain model from a steady-state calculation to a transient one. Without modeling the full continent you need to impart information to this regional model about how ice-thickness changes and ice-flow changes inside and outside of this domain correspond to changes in the rest of the ice sheet in which it is embedded. So, for the goal of regional modeling you are minimizing even more assumptions (and challenges in setup and computation) by starting in steady state. I looked at this for 2.5-D (flowband) models: Koutnik and Waddington (2012), Well-posed boundary conditions for limited-domain models of transient ice flow near an ice divide, Journal of Glaciology 58, 1008-1020.**

Yes, in fact we are meeting 2 different problems that are linked: - Solving the thermomechanical problem needs a lot of time (even on a restricted area) so that the thermal state first reaches a certain stationnary state that can be used as a starting point. - Then, solving the time-dependant problem on a domain limited by virtual boundaries

is very tricky as it needs to permanently conserving the mass out of the domain, while maintaining the global shape of the dome. To do so, the parameterization of the boundary conditions is very sensitive and would need specific developement.

That's why we decided to make things simpler, and to separate the problems: 1 estimating as best as possible the melt rate (which is why the ice thermal state really needs to be described), and this was done in a previous paper (Passalacqua et al 2017) and 2- considering these ice temperature and melt rate as true, and dealing with ice mechanics, to see the influence of the bedrock description (this paper).

L113: We completed the end of the paragraph: "Solving the coupled thermo-mechanical equations would require excessive computing resources, without radically changing the ice fluidity – which is mainly controlled by temperature. Similarly, we do not account here for long-term evolutions of the ice sheet surface, but are aware that this assumption strongly affects the trajectories of the ice particles."

**Line 94: Instead of 'more influent', suggest 'has more influence'**

L122: Changed for "where shearing has more influence"

**Line 110: State as 1/lambda?**

L137:Changed for "The age resolution is stated as 1/lambda"

**Line 113-114: This sentence wasn't clear to me starting with 'The way ice strains. . .'**

L139: The couple of sentence was modified, to say that a given ice flow depends on the bedrock underneath, but a given bedrock can lead to different ice flows: "The way ice strains by flowing over a rough bed differs depending on the shape of the bedrock underneath. Similarly, a given bedrock shape can be a convergence or a divergence area, depending on the orientation of horizontal ice flow."

**Line 143: What do you mean by 'a logic combination'? L168: Changed for "boolean combination".**

**Figure 2: x-axis and y-axis numeric labels are way too small in the top panels, and probably also too small in the bottom panel. Axis labels are missing. I am not really sure where to read the numbers from each panel and without those they don't say much – the caption needs to be improved to make sense of these panels, or maybe the top four are not shown? It looks like there would be more overlapping areas, or is it just that age resolution is limited? Make sure the caption clearly guides the reader through this and that all box colors are identifiable. For example, it took me awhile to see the blue box showing location of Van Liefferinge results. Seems like colored crosses should receive more discussion in the text. Pros / cons of each choice could be takeaway points. Again, how should the community use these results?**

Numeric labels on top panels were simply deleted, as they are identical to the ones of the bottom panel, and axis label was added. Indeed, age resolution is the most restrictive parameter of the 5 criteria. The selected areas are now colored in yellow, which is more visible. We added discussion on the sites in the text: "Considering that, only a few set of favourable drill sites remain in boxes A, B and C (blue, orange, red and yellow points in Fig2 2. Red and blue points have less risks of basal melting, while yellow and orange have less risks of stratigraphic disturbances. The best choice between these sites should be now guided by local radar surveys characterizing the internal layering of the ice, and the vertical strain rate profile (Nicholls et al, 2015)."

**Putting these results in better context with the Van Liefferinge et al. (2018) results seems necessary. Especially that this work is still in The Cryopshere Discussion, the reader is not necessarily sure what to make of these parallel efforts. I might have missed it, but I think the first citation of this work is in the caption of Figure 2. Section 3.5 goes into more depth between the approaches but could be worthwhile to put this context up front, and as part of the framing of why your results matter to the community.**

The presentation of VL et al (2018)'s work is shifted to the end of the introduction, and is followed by explanations on what should be done with these data:

L77: "The decision-making process of a drill site needs both field survey and modelling, the former feeding the latter with geophysical constraints, and the latter reducing the areas of interest for new field surveys, focusing more and more on promising sites. The information brought by the present study should be sufficient for a last dense radar survey to be led on promising sites during the next field season, at a scale of a few hundred of meters. Then the community should be able to take a decision for a drill site in the Dome C region."

**Is there really no way and/or no effort underway to combine these two approaches? Is the limitation only computational? At what point might this be possible? (Or, what are the next steps that can be taken by the modeling community in this effort to find the best drill site?)**

Of course we could go further in ice modelling, for example by using VL (2018) probabilities that ice reached the melting point in the last 1,5 Ma as a boundary condition in the 3D model. But we are not sure it would necessary be a good idea:

- Having several points of view give a supplementary information on the strength and weaknesses of each approach.

- At the scale of a few hundreds of meters and less, our choices should be guided by the careful analysis of the future local observation data, not new modelling.

**Line 182: I suggest rephrasing 'oldest-ice challenge' – even 'the challenge of finding the oldest ice' sounds better, somehow the other sounds too loose, and therefore does not have as much impact.**

L206:We changed for a simple formulation "This zone of higher shear should be discarded for a future drill site".

**Line 187: 'appropriate ice', suggest rephrasing what you mean by 'appropriate' or using the phrase 'candidate drill sites'**

L210: "The best combination of age, age resolution, folding, convergence and melting criteria is shown in Fig. 2 (bottom), revealing several spots of ice reaching the 5 criteria."

**Line 190: suggest using another word than 'risks' (there are other usages that I'd also change'; maybe the word is 'chance'**

L225: "The ice flow converge, increasing the possibility of insufficient age and positive basal melt"

**Line 190: By shorter trajectories you really just want to be closest to the dome / ridge-**

L214: Indeed, we now specify this point: "Locations within boxes C and A are closer to the ice ridge and should be considered first for a future Oldest Ice drilling because of shorter trajectories."

**Figure 3: Why not give the x-axis in km? All labels are very small Is the bed topography the most important boundary condition in your modeling? What resolution would be ideal to be confident from modeling on where to pick a drill location?**

The x-axis is now given in km, and labels are bigger. As discuted in the text, the boundary condition is crucial for the basal age and age resolution, and the shear history of the ice. That is why this figure shows the amplitude of the bedrock spatial undulations. Given the underlying assumptions of this kind of work, improving the spatial resolution from 1 km to 500 m or 100 m would not weaken these assumptions. A better bedrock description at the scale of a few hundred of meters is required locally, but, at this scale, the 3D ice flow modelling will not necessary be a better tool than a fine interpretation of the radargramms and their internal layers.

**Line 210: How can these results really be benchmarked against Van Liefferinge et al. (2018) given that they use a very different approach? All you have done is qualitative comparison, right?**

Maybe the word "benchmarked" is less appropriate than the word "compared" in this case. This comparison is made by overlapping the results of the two approaches. Any

model is as good as the assumptions on which it is based, and comparing the two approaches is a way to have two different points of view on the same object (the future drill site, maybe where the model agree), but also to have information on the strengths and weaknesses of each model (where they disagree).

**Line 214: should be 'constraint'**

L73:Changed for "constraint"
* * *

---

## Author Comment (AC2) · 18 May 2018

This manuscript uses a three dimensional ice sheet modelling approach to explore the basal age and resolution of the ice sheet at Dome C, in a region where bedrock relief is likely to be conducive to preservation of very old ice: up to 1.5 Ma or more. The study appears well-posed and the work is a nice summation of what this approach can tell us regarding basal age and resolution. It is an important advance that is required for targeting future drilling locations and should be valuable in guiding additional exploratory studies. I have only minor comment concerning the modelling itself. The paper does suffer in places from somewhat non-standard English usage, some of which intrudes a

little on readability. While fully appreciative of the authors' first language I respectfully suggest these items be edited for clarity – noting a native English speaking author is on the list. I see the other referee has commented in similar vein – I will not generally specify the linguistic items for correction below.

We would like to thank Tas Van Ommen for his fruitfull comments, that helped improve this manuscript. English wording was carefully corrected in this new version.

**Detailed comments: Title: In general I disfavour the usage of Dome C as a "point location" synonymous with Concordia Station, as the entire region is really Dome C. Suggested use would be to have the title read "Oldest Ice" patches diagnosed at Dome C, 37 km southwest of Concordia Station. I however leave it to the authors to consider, as it is not a substantive concern.**

Now we changed "Dome C" for "the Dome C summit", when we refer to the Dome C upper point.

**Line 25: Define IGE on first use**

L28: The acronym is now developped "IGE (Institut des géosciences de l'environnement, Grenoble)"

**Line 37: "and [provide] sufficient resolution"**

L42: "The observed isochrones are compatible with high basal age and provide sufficient resolution"

**Line 48: Fischer et al. actually stipulate no more than 20 ka m-1 although this may now be thought too coarse. I have heard targets of 14 ka m-1 used. In any case, the 10 ka m-1 is not consistent with the reference.**

This reference was indeed an error. This threshold value was given by J. Chappellaz, following the different Beyond EPICA-Oldest Ice meetings and workshops.

**Figure 1 caption: "show the hold [sic] of the domain"**

"The red patch on the context map shows the domain used for the calculation"

**Line 74: relaxed for 50 years . . .. How is the reader assured that this is adequate? Naively it seems very short. Maybe just reword to say that this period proves sufficient to propagate away initial discontinuities or similar.**

Ideally, we should have reached a steady state. But reaching a true steady state for this 3D model with virtual boundary conditions is a very challenging task (ensure mass conservation, limit BC influence on the surface shape, control of the shape of the ice surface). So this relaxation time is a trade off between a fixed geometry simulation (which can lead to misinterpret the results if the surface shape is not consistent with the flow law), and complete free-surface simulation, which would require much more developments. Since the goal of the study is a comparison in-between of different sites, this question of transient vs steady simulations is less crucial than it could be.

We added the following sentence: L99: "The ice surface is very flat, and 50 years is enough to accomodate the surface altitude to the ice rheology up to âĹij 1 m, without radically changing the orientation of the ice ridge, on which we have little control."

**Line 77: "focus on a region where basal melting is probably null" - this may be true for the high points, but the domain most certainly includes areas of basal melt, so how is it that a no sliding condition is OK?**

A no sliding condition is valid because the horizontal velocities are very small, so that even if there is sliding, the sliding component of the horizontal velocity would be even smaller. We completed as follows: L103: "We here focus on a region where basal melting is probably not present or limited, and horizontal velocities are very small, so that, for the sake of simplicity, a no-sliding condition is imposed at the bottom of the ice column."

**Line 95-100: It is not clear from the description why the use of stress exponent n=3 is valid. Indeed there is some varied opinion in the literature over the best value to**

use in various situations particularly ice divides (see e.g. Martin et al., JGR, 2009; Martin and Gudmundsson,TC, 2012; Petit et al., JGlac, 2007). While not wishing to create imbalance in the treatment in this paper by opening an extensive discussion, some context to the literature would be useful. More importantly for understanding the results of this modelling, could the authors arrive at a statement as to whether an exponent n=3 is likely to under- or over-estimate age and resolution? That is, is it conservative to the aim of finding old ice?

The question of the value of n is a specific scientific question, and its complete discussion largely exceeds the goal of this paper. Even if we know that its value is a matter of debate, we first made sure that the value of n was compatible with observed velocities, which should be sufficient for our purpose, even if it also influences deep layers. Our goal is not to make the best evaluation of the absolute age (which is much more influenced by a bit of basal melting than by a change in Glen exponent), but to compare some locations to others. For this purpose, changing the value of n would not radically change our results.

**Lines 128-130: Maybe an example of the language clarity issue, but it is hard to see what is meant by "the outputs still keep their relevance when analysed relatively to themselves"**

We changed for the following sentence: "If the absolute value of the age, age resolution, or strain rates can be discussed regarding the choices of the model parameters, we reckon their spatial variabilities are robust since they mainly depend on the shape of the bedrock and of the ice surface"

**Line 138: the "water limit" at 480m needs a little explanation, where does it come from and what is the reference height (I assume it means 480 m.a.s.l.).**

L156: We add a sentence: "We will call this threshold "water limit", above which there is no evidence of the presence of water in the radargramms"

**Line 171: An example where the "Dome C" not equal to "Concordia" nomenclature issue comes up. I'd favour "Concordia".**

Changed for "the Dome C summit"

**Figures 2 and 3: Axis labels in particular are too small. Figure 3 would benefit from all text being larger.**

Axis label have been enlarged, or deleted on the upper panels as they are identical as the ones of the bottom panel. Text is larger in Figure 3.

**Line 216: "our biggest central patch" isn't so easy to follow as using the labels provided: I assume it is "Patch A".**

L235: Changed for "box A"

---

## Author Comment (AC3) · 18 May 2018

**The manuscript presents a potentially valuable 3D modelling-based study designed to isolate, through a series of spatial masks, candidate locations for 'oldest ice' (1.5 Ma) near Dome C, East Antarctica. The analysis is interesting and, I believe, robust (subject to some reservations, below) and I would support publication. I believe the manuscript structure and approach are valid, but I do see the manuscript's current findings as somewhat undermined by the handling of temperature – and particularly basal temperature and sliding - in the analysis. I would encourage a revised manuscript to consider this in more detail, at least placing some first order approximations of error**

based on possible temperature scenarios. I accept that this may not be the forum for a full thermo-mechanical analysis but, for the analysis is fit for purpose, it needs to report some approximation of the age errors that might derive from the assumptions made. The writing is occasionally ambiguous and includes grammatical and typographical errors.

We would like to thank Bryn Hubbard for his fruitfull comments on this manuscript. We understand your concerns, as this drill site location needs many glaciological aspects to be discussed. However,we think that handling the problem in a comprehensive way (ideally: 3D transient simulations assimilating radar isochrone layers, basal reflectivites and surface velocities, and uncertainties discussion) is far too ambitious. That is why we handled the problem differently. The age estimation was first done by Parrenin et al (2017), using dated internal layers. Similarly, the influence of the geothemral flux is crucial, and its local value and influence on long term is discussed in Passalacqua et al (2017). Given these two previous studies, we only discuss here one single simulation, which is enough to compare the local glaciological properties given a certain bed shape. The estimation of uncertainties would require sensitivity tests on several parameters (fluidity, Glen exponent, temperature profil/basal melt rate), which is very heavy to do in 3D. In fact, the goal of this paper is not to assess the best estimation of the age and its uncertainties. To make this more clear, we added a sentence to precise this point for the reader:

L51: "Note that the goal of this study is not to assess the best age estimation, but to evaluate which sites have better glaciological properties than others."

**Some more specific comments follow:**

Some of the wording could be improved here between lines 3 and 7.

The abstract was deeply reshaped:

L5: "A 3D ice flow simulation is used to calculate five selection criteria, which spatial

variability is used to locate areas that have better glaciological properties than else-where. These selected areas (a few km 2 ) lie on the flanks of the Dome C bedrock high, where a balance is found between risks of basal melting and sufficient age reso-lution."

**Page 2 There are at least two typographical errors on this page ('serie' and 'de-favourable'). The manuscript needs checking to remove other occurrences. I identify a few more below.**

We changed for "set" and "unfavourable"

**30-31 Is it not possible that changes in ice thickness have an influence here at long timescales? Can this influence and outcome be approximated?**

Indeed, and this previous results (Passalacqua et al 2017) already accounted for the ice thickness changes. We here use a melt rate averaged over the last 400 000 yrs.

**30-42 Can the simulations and models of others publishing on this topic be sum-marised briefly?**

We restated or added precision on these two models.

L31: "First, a 1D heat model was run over the last 0.8 Ma to determine the present state (temperate or cold) of basal ice, which was compared to the reflectivity map in the region of Dome C. We could infer the value of the local geothermal flux, and explain} the origin of the local spatial distribution of subglacial water at the ice-bedrock interface. (. . .) Second, a kinematic 1D ice flow model was used to evaluate the age and age resolution of the deepest portion of the ice sheet. The distance between the dated isochrones and the modelled ones was minimized to infer a thinning parameter that characterizes the vertical deformation through the ice column."

**77-80 This reads as contradictory, including a statement that 'basal melting is proba-bly null' (I would use zero rather than null) and 'Vertical velocities. . . are equal to the basal melt rate output from previous modelling. . .'. I think the manuscript would benefit**

from an explanation and statement of possible error (in using this basal temperature field) here.

If the melt rate is null, the no-sliding condition is satisfactory. If there is a bit of melting, as the horizontal velocities are small, the sliding part of the horizontal velocities would be even smaller. But there is no contradiction with the vertical velocities being equal to the melt rate. So we modified the first sentence as follows: L103: "We here focus on a region where basal melting is probably not present or limited, and horizontal velocities are very small, so that for the sake of simplicity, a no-sliding condition is imposed at the bottom of the ice column."

**94 'has more influence'**

L122: Changed for "has more influence"

**124 'used'**

L152: Changed for "used"

**128-130 This sentence is unclear 132-133 I do not believe this is a valid argument: either the analysis is fit for purpose or it is not. For the former to hold then errors need to be constrained. If, as a consequence of the 'practicable' analysis undertaken, errors are large then the manuscript would benefit from those large errors being stated as usefully as possible.**

This sentence was unwise (128-130). The idea is that our approach is more a comparison between sites rather than a discussion on the value of the age. The paragraph is restated as follows: L156: "If the absolute value of the age, age resolution, or strain rates can be discussed regarding the choices of the model parameters, we reckon their spatial variabilities are robust since they depend mainly on the shape of the bedrock and of the ice surface. As a consequence, this study focuses on comparing promising locations the ones with the others rather than on discussing the influence of model parameters."

**158-159 Presumably reflected radar power could inform as to the current location of this boundary. Do such data exist and/or has such an interpretation been published elsewhere?**

The problem with the bedrock reflectivity as a function of bed elevation is that that needs the ice attenuation, which we generally calculate from correlating bed echo strength against bed elevation (eg Zirrizotti 2012)... which is circular. So we are confident that our approach is the simplest one and the more unambiguous.

**170 'arch' and 'radargrams'**

L195: Correction made

**Figure 2 Given the importance of these domains I find that the green on brown shading isn't working very well. In fact, this figure could be improved in several ways including: formally numbering panels a-e; increasing axis label font size in a-d; changing the depiction of the oldest ice targets considering the underlying green bed elevation band; and rewording the first line of the caption to a more standard format.**

We changed the color of the selected areas for a yellow one, and we labelled the upper panels. Axis label of upper front are deleted since they are identical to the ones of the bottom panel.

**183 I wonder why this is 'discarded' rather than included as a mask in the way that other spatially-distributed variables are?**

We could have added a specific mask for "ice that crossed this high-shear zone", but no significant area concerned remained in the combination of the 5 masks, so it was not necessary. Anyway, a basic reasoning on the length of the trajectories is enough to focus on areas upstream of this high shear zone. Now we highlight the higher-shear zone by a specific mattern on the map.

**186 This reference to 'bottom' highlights the need for panel labelling in the figure**
L211: We now refer to panel e)

**200 'However, it allows a restricted area to be defined where. . .'**

L224: Changed for "However, it allows a restricted area to be defined where a new set of observations will be the most valuable"

**206 I believe 'overhanging its environment' is inaccurate and does not convey the intended meaning. I think this argument needs to be clarified and formalized.**

The argument needs a figure to be much simply explained, and is anyway not crucial. So we removed this couple of sentences.

**Figure 3 caption 'shows' Correction made**

**221 I would replace 'On the contrary' with 'In contrast'**

L240: Changed for "in contrast"

**223 Could the references here to 'upper part' and 'left part' be replaced with compass directions north and west?**

We could do that, but as the north is not oriented upwards, indications of orientations are not intuitive at all, so we keep the present formulation.

**226 I'm not familiar with the third from final word in the sentence.**

L245: Changed for "criterion"

---

## Editor Comment (EC1) · J.-L. Tison (Editor) · 23 May 2018

Dear Authors,

We have now received the comments of the three reviewers, and you have answered individually to each of them.

Reading these, I believe you can work on a revised version of the manuscript. It will obviously require improvement of the written english, as commented by the three reviewers. Please use the skills of your co-authors that are english-natives!

To ease the procedure on my side, I would recommend you produce three documents:

a) a synthesis document of your replies to the three reviewers comments point by point, referring to changes in your revised manuscript (line numbers referring to the new manuscript with highlighted changes (see b))

b) a version of the new manuscript with highlighted changes and line numbers

c) a "clean" version of the new manuscript

Good luck with it,

Best regards,

Jean-Louis Tison